# Gaining a Sense of Touch Object Stiffness Estimation Using a Soft Gripper and Neural Networks

Michal Bednarek * , Piotr Kicki , Jakub Bednarek and Krzysztof Walas 

Institute of Robotics and Machine Intelligence, Poznan University of Technology, 60-965 Poznan, Poland; piotr.kicki@put.poznan.pl (P.K.); jakub.bednarek@put.poznan.pl (J.B.); krzysztof.walas@put.poznan.pl (K.W.)
* Correspondence: michal.bednarek@put.poznan.pl

**Abstract:** Soft grippers are gaining significant attention in the manipulation of elastic objects, where it is required to handle soft and unstructured objects, which are vulnerable to deformations. The crucial problem is to estimate the physical parameters of a squeezed object to adjust the manipulation procedure, which poses a significant challenge. The research on physical parameters estimation using deep learning algorithms on measurements from direct interaction with objects using robotic grippers is scarce. In our work, we proposed a trainable system which performs the regression of an object stiffness coefficient from the signals registered during the interaction of the gripper with the object. First, using the physics simulation environment, we performed extensive experiments to validate our approach. Afterwards, we prepared a system that works in a real-world scenario with real data. Our learned system can reliably estimate the stiffness of an object, using the Yale OpenHand soft gripper, based on readings from Inertial Measurement Units (IMUs) attached to the fingers of the gripper. Additionally, during the experiments, we prepared three datasets of IMU readings gathered while squeezing the objects—two created in the simulation environment and one composed of real data. The dataset is the contribution to the community providing the way for developing and validating new approaches in the growing field of soft manipulation.

**Keywords:** machine learning; tactile sensing; perception for grasping

## 1. Introduction

Humans have an innate ability to perceive the physics of the world around them. As we are biologically equipped with a very sophisticated sensory system that delivers data to the brain, no-one consciously plans how to grab a cup of tea, squeeze a wet sponge or flip a book page. We all know how to do that and how to predict deformations of different objects based on their physical properties. Moreover, humans have at their disposal soft and highly effective grippers—hands. Taking into account our assumptions about the world that come from our minds, combined with the embodied intelligence [1] of our hands, we can flawlessly adjust the process of manipulation to fluctuating external conditions. However, machines do not have such in-built proficiency. Thus, their ability to manipulate only allows for handling repetitive tasks and prevents them from adapting to new types of objects efficiently.

Biologically inspired soft grippers [2–5] are designed to handle not only rigid bodies but also deformable and usually delicate objects. How they interact with the real world and how they adjust to different objects is ruled by their property called *intelligence by mechanics* [1]. One can observe a significant rise in the number of available applications of sensors capable of capturing high-dimensional deformations of soft and unpredictable physical objects [6–8]. However, in our work, we state that traditional and widespread sensors based on microelectromechanical systems can also be successfully used to predict the physical nature of the robot's surroundings. Thereby, we propose a hybrid approach that connects an *embodied intelligence* of a soft gripper with an *artificial intelligence* system to

provide an easy to use, open-source and inexpensive method of estimating the physical properties of objects with various stiffness parameters.

The following study presents the deep learning, real-world application for stiffness coefficient estimation based on data from Inertial Measurement Units (IMUs) attached to the fingers of the gripper. Our contributions are:

1. Creation of simulated environments for generating contact signals from IMU and examining the soft gripper in various scenarios.
2. Verification of the performance of three neural networks in the task of stiffness parameter estimation—one purely convolutional and two recurrent models.
3. The real-world verification of the proposed solution.
4. The extensive examination of the reality-gap between the simulated and real data.
5. The open-source implementation and data used in the experiments available online (https://github.com/mbed92/soft-grip).

To prepare the real-world experiment, we used a two-finger gripper based on the Yale OpenHand Project [2] with two IMUs attached to its fingers. The motivation behind the choice of that type of sensor is twofold. First of all, typically soft grippers have no hinges and do not use encoders; therefore, we cannot track their movement directly. Following the research on the Pisa/IIT SoftHand [9], the the IMU measurements are sufficient for the motion tracking of underactuated and elastic fingers of the gripper. Secondly, IMUs are inexpensive, small and widespread among the robotics community. We than replicated this setup in simulation to obtain more learning data and the control over generated signals.

The course of the research is shown in Figure 1 and consists of the following stages: first, for the set of deformable objects of different shape and different stiffness parameters we performed squeezing motion both in simulation and in the real world. In both cases we were registering IMU data. For both approaches we trained and tested three different neural networks architecture. The final outcome of the learning process were estimated/regressed stiffness parameters of the objects. We started our investigation with experiments carried out exclusively on data from the physics simulator to verify the capabilities of three different architectures and examine the generalisation of the stiffness parameter regression between different shapes of squeezed objects. Thereafter, we investigated the problem of closing the reality gap between the simulation and real-world data. In our experiments, we exploited the MuJoCo [10] simulator to provide a sufficient number of training samples. The IMU device model used in our work was the MPU-9250 model. In Figure 2, there is presented the setup used in the real-world scenario with its simulation model and exemplary objects.

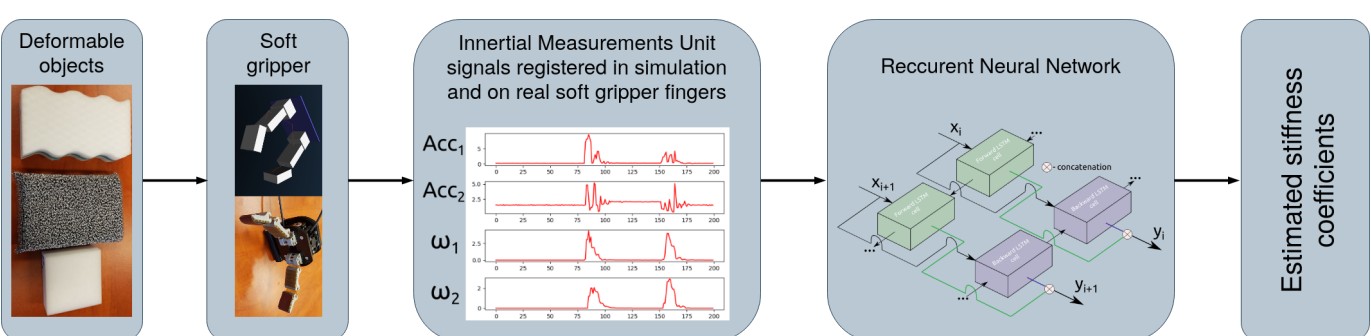

**Figure 1.** Schematic diagram of the proposed system. The set of deformable objects is squeezed in simulation and with the real gripper. The data is registered from Inertial Measurements Units mounted on the gripper fingers. Recurrent Neural Network is performing deformable object stiffness coefficient regression based on registered data from IMU.

The remainder of the paper organised as follows. First, we will review the state-of-the-art in the field of physical parameters estimation from haptic data. Then, we will provide a description of prepared setups and our experiments. Next, we will move on to the results section followed by the discussion. Finally, concluding remarks will be provided.

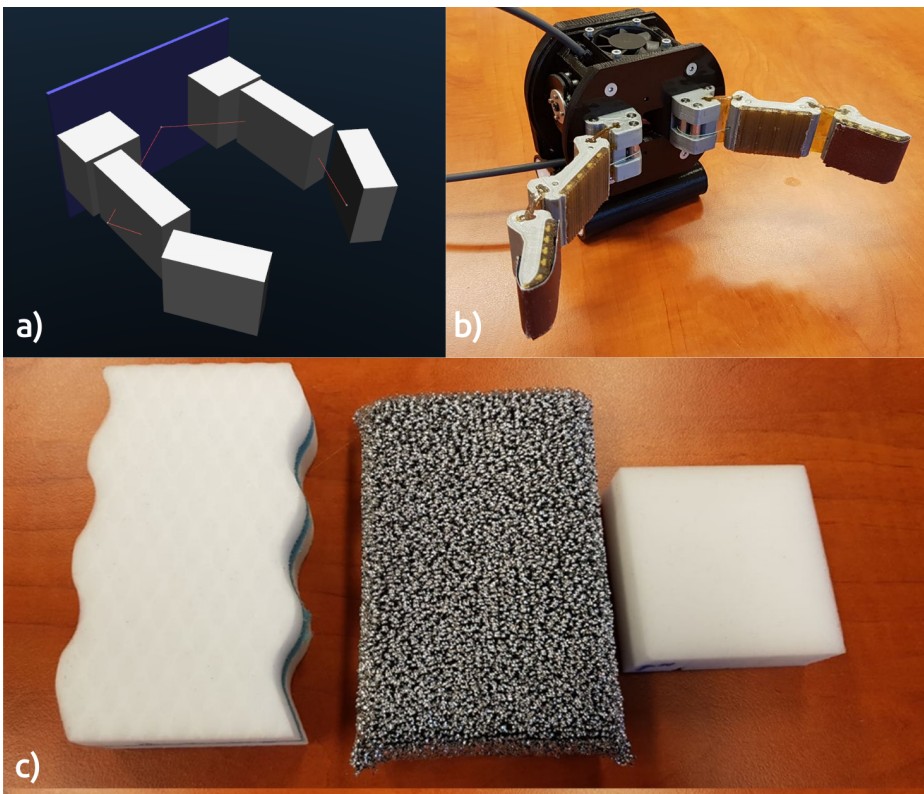

**Figure 2.** To test our system, we arranged a real-world scenario using a 2-finger Yale OpenHand gripper [2]. To provide a sufficient number of training samples for the learning process, we modelled the gripper in the MuJoCo simulator as it is depicted in (**a**). In (**b**), real fingers consist of three plastic blocks with flexible parts made of urethane. In (**c**), there are presented examples of sponges, exposing different stiffness, used in our real-world experiments.

## 2. Related Work

In this section, we provided a comprehensive literature review both on the approaches to measuring and estimating the object stiffness. Further, we showed current advances in processing data from IMUs for a wide range of purposes. Finally, we presented a brief overview of underactuated grippers with an emphasis on the soft grippers.

### 2.1. Measuring and Estimating a Stiffness

The knowledge about material's stiffness is highly demanded in many practical applications such as industrial robotics, where a robot may use this information to predict an object's deformation. We present current advances in finding object stiffness in two general approaches: *measurement*, where the result was obtained with the usage of advanced, dedicated sensors and *estimation*, where we focused on the possible use of all available information relevant for a given task.

Measurement—The practical application of the stiffness measurement was shown in [11], where the authors proposed a method for continuous rail rigidity measurements using the accelerometer and oscillating mass on the rolling wheel. This indicates that the issue under examination is of great importance in engineering. Unlike the previous method, the noncontact measurement of spindle stiffness was presented in [12]. The authors proposed a magnetic loading device that enables one to perform the measurement while the spindle rotates. Due to the usage of magnetic loading, that method is limited to the ferromagnetic objects. Measuring the stiffness is also possible at a much smaller scale. The authors of [13] presented the review of the nanoindentation continuous stiffness measurement technique and its applications. The range of stiffness coefficients of materials is extensive. To avoid saturation and enhance precision, authors of [14] proposed a portable measurement tool able to adjust the sensing range by manipulating tool parameters, such as

touch module separation, indenter protrusion and spring constant of the force sensing module. Authors of [15,16] analysed the stiffness measurement techniques applied to the polymer foams, which are cognate to those used in this paper. In [15] a procedure for measuring the stiffness of the object using dot markers on the object and compression plates to exert the force on the object was proposed. Authors stress the fact that nonaxial compression tests result in worse performance, but it is usually the case in robotic manipulation. As in our method, in [9] authors proposed the IMU-based approach but in a different task—the reconstruction of the configuration of a soft gripper. As opposed to that work, we propose to indirectly measure the stiffness property by the change of behaviour of the soft gripper while squeezing, not the gripper's configuration itself.

Estimation—The method for the object stiffness estimation from the force sensor readings was proposed in [17]. Authors used small optical force sensors mounted on the fingertips, a known kinematic model of the robotic hand and a vision system to calculate the stiffness based on the force and displacement readings. An alternative approach that does not require measuring the object deformation was proposed in [18]. The authors proposed the Candidate Observer-Based Algorithm, which exploits two force observers, with different stiffness candidates, for estimating the stiffness of objects with complicated geometry. Unfortunately, the authors did not refer their method to the ground truth stiffness measurements. However, such a comparison was made in [19], where the neural network was trained to predict the stiffness coefficient based on the maximum penetration and the maximum contact pressure variation. An alternative deep learning approach for understanding the haptic properties of objects was proposed in [20]. The real-world objects were classified in the set of haptic adjectives in the multilabel fashion based on haptic signals from BioTac sensors [21] and images. That work shows that there exists a correlation between haptic sensor readings and the structure of the real-world objects, and in our work, we took advantage of that fact.

The extensive overview of machine learning methods in the soft robotics aspect is described in [22]. In the context of sensing, the authors distinguish sensor characterization and systems characterization. In the group of sensors characterization, the use of Recurrent Neural Networks for parameters regression is widespread, as we are dealing with signals and continuous values of sensor parameters. On the system characterization level, we are more focused on higher-level labels successful grasp [23] or slip detection. The use of the classification of signals with categorical values is more common. The more focused approaches are shown in [24], where learned control mechanisms were used, reinforcement learning [25] or learned differentiable models [26].

### 2.2. IMU Measurements Applications

The popularity of IMU usage stems from its widespread availability at a low price. One possible use in the robotics community is a robot's state estimation. In [27], acceleration and angular velocities collected from sensors located on the humanoid leg, together with joints positions were used to estimate the velocity of the robot links. Authors in [28] presented multiple interesting approaches to measure the ground reaction forces indirectly during the human walk with the use of wearable IMUs. The other field where the measurements of acceleration can be utilised is a material classification. In [29] authors used the haptic device SensAble Phantom Omni [30] to gather the accelerations and velocities while scratching the material surfaces. That dataset was used in [31], where a deep convolutional neural network was taught to map raw signals input to classes of textures. The presented method stays close to our solution. However, in our work, we performed regression instead of classification.

### 2.3. Underactuated and Soft Grippers

As the approach proposed in this paper requires a soft gripper, we present a brief overview of existing underactuated and soft grippers.

Underactuated grippers was an area of research for many years. One of the first grippers was designed by Tomovic and Boni in [32]. They proposed an anthropomorphic underactuated hand with five fingers and 14 joints, driven by a tendon-driven mechanism and two servo motors. Using tendons in the gripper designs provides an ability to easily adjust the gripper's shape to the object in a gentle way. To achieve grasps available only to the fully-actuated mechanisms, authors of [33] proposed an underactuated gripper with electrostatic brakes in joints, which enable to carry heavy objects by reducing power consumption and motor torque during a steady grasp. Nowadays, hybrid approaches, which combines fully actuated fingers for precise manipulation and underactuated ones for power grasp and compliance, are becoming popular [34,35]. A different hybrid approach is presented in [36], where authors proposed an underactuated gripper with a suction cup for picking up various objects in different working environments. Such hybrid solution allows for building multifunction robotic cells [37] for maximising the production rate.

On the other hand, for full compliance and shape adaptability, there is a lot of research about a special group of underactuated grippers—soft grippers [38]. A popular way of designing soft grippers is using elastomer actuators. In [39] authors used rubber fingers driven by the pressure in the chambers located inside the fingers. However, as used materials are usually soft, they are susceptible to damage. In response to that, authors of [40] presented usage of self-healing materials to construct a soft gripper able to repair itself. A different approach to control soft grippers is to use dielectric [41] or shape-memory based [42] actuators. However, probably the most popular group of soft grippers are those with passive structure driven by the external motors, such as adaptive compliant gripper proposed in [43] or biomimetic soft-hand [44]. To this group also belongs a Yale Hand gripper [2], which we used in our research. It is a low-cost two-fingered open-source underactuated robotic hand, which is built with 3D-printed components with compliant, flexible joints, and driven by tendons actuated with servos.

Interested readers may refer to recent more comprehensive underactuated and soft grippers reviews such as [34,38,45].

## 3. Method

In the following section, we described the experimental design and provided detailed information about both real-world and simulated environments for our experiments. Then, we described the proposed neural network architectures used in our research.

### 3.1. Real Data

The Yale OpenHand shown in Figure 2 is the underactuated, two-finger soft gripper with joints in the form of urethane elements to assure the elasticity of fingers. The real-world model was 3D printed and driven by hobby servos capable of generating a force up to 10 N. The IMUs were mounted at the fingertips of the hand. The IMU readings were used to estimate grasped objects stiffness. In our work, we assessed how the embodied intelligence of such soft gripper could be used alongside with the artificial intelligence system to predict the real stiffness coefficient of a squeezed object. In the following paragraph, the real-world data gathering process was presented.

First we estimated the stiffness coefficient for real objects in the dataset. To calculate ground-truth values of the stiffness coefficient of real-world objects, we used the Universal Robot UR3 collaborative manipulator, which was able to measure torques and forces in its joints and tool respectively. The robot had 3D printed plastic bar mounted at the flange. Using the Dynamic Force Control mode and pressing objects with the desired force, we were able to accurately measure the displacement under specific force from robot state readings. Thus, the stiffness parameter was computed according to Equation (1), where $f_1$ and $f_2$ are forces in Z-axis while pressing an object with a tool and $|d_1 - d_2|$ is the relative distance that correspond to the deformations under $f_1$ and $f_2$. We are aware that chosen objects express nonlinear behaviour in their stiffness model (e.g., the greater robot compress the sponge, the less deformation it adds). However, objects did not reflect that nonlinear

effects in the specified range of exerted forces. Therefore, in our work, we assumed that the estimated stiffness parameter is homogeneous for the entire object. Table 1 contains stiffness coefficients measured experimentally for each object.

$$k = \frac{|f_1 - f_2|}{|d_1 - d_2|} \tag{1}$$

**Table 1.** Stiffness coefficients computed for 5 different real objects.

| Object | Stiffness [N/m] |
| --- | --- |
| Wire sponge | 909 |
| Hard sponge | 1020 |
| Polish sponge | 735 |
| Soft sponge | 380 |
| Squash ball | 1353 |

After measuring the value of the ground-truth stiffness coefficients, we used Yale OpenHand to perform squeezing motion of each object and collected IMU readings during motion execution. In total, we gathered 500 series. They consist of 12 sensor readings ($2 \cdot$ IMU readings: $[a_x, a_y, a_z, \omega_x, \omega_y, \omega_z]$) each 200 time steps long. All samples are equally distributed among the objects–100 samples per each object. The data was split into two subsets—200 train and 300 test samples that were used in sim-to-real experiments. Both sets in all our experiments remain unchanged. Thus, test data is never used in the NN training. To address the issue of a physical interpretation of obtained stiffness, taking as input the accelerations and angular velocities, we claim that the motion of gripper fingers registered while squeezing different objects would vary significantly, which was presented in Figure 3. One can observe that depending on the object's stiffness the magnitude and oscillations of both-angular velocity $\omega$ and linear accelerations *Acc* were significantly different from each other, e.g., in the range of values or the oscillation rate. Taking that phenomenon into account we put forward the thesis that it is possible to distinguish between different stiffness parameters in the space of IMU sensors registered during squeezing of these objects.

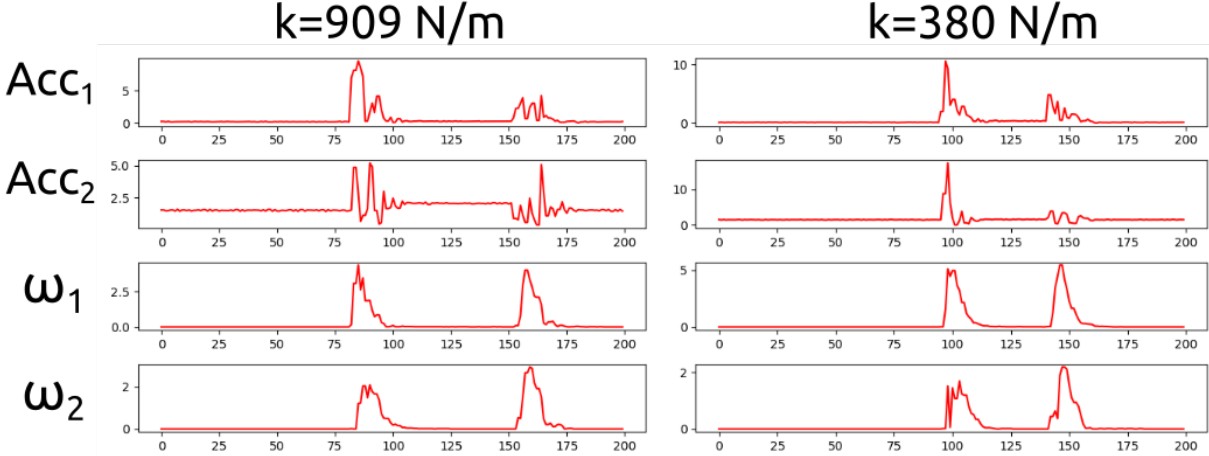

**Figure 3.** Comparison between exemplary samples from the real-world dataset while squeezing objects with different stiffness values with a soft gripper. Values $a_1$, $a_2$, $\omega_1$ and $\omega_2$ refers to the magnitude of registered accelerations and angular velocities and are expressed in $\frac{m}{s^2}$ and $\frac{rad}{s}$ respectively.

However, in our real-world dataset, there were only five different objects with distinct stiffness values, which served as labels. In that situation, the number of different labels was not sufficient to perform a successful regression. In fact with such a low diversity,

a regression would inevitably turn into a classification and that was not desired in the task of stiffness estimation. To overcome that problem, we prepared a second dataset based on the simulation, where there was a possibility to generate more training samples. Stiffness coefficients were adjusted to meet measured values.

### 3.2. Simulation

Modern neural networks frequently suffer from the limited ability to generalise to new domains which are out of their training dataset. However, the rising popularity of machine learning techniques in the robotics community leads to a significantly increased need for data from a variety of experiments. To fulfil that demand, the state-of-the-art approach is to perform experiments in simulation and use them to feed neural networks. In the case of tasks which involve physical interactions, researchers can choose from a wide range of available physics simulators. In our case, we selected MuJoCo physics simulator, due to its new features regarding soft objects modelling. The simulated soft-robotic gripper was shown in Figure 4. Fingers were connected by tendons and they are pulled by the actuator, which simulates the pneumatic cylinder. Our model was based on the three-finger real gripper [3] but with one finger removed. As it is depicted in Figure 4a, during experiments, our gripper squeezed and released objects of three shapes—a ball, box and a cylinder, all with a variable stiffness parameter. To simulate elastic deformations of the gripper, each geometrical block of each finger is connected to others by three hinges. In this setup, we can easily adjust the ranges of each joint in a roll, pitch and yaw axes, as was depicted in Figure 4b. Finally, each 8-block finger behaves similarly to the elastic finger.

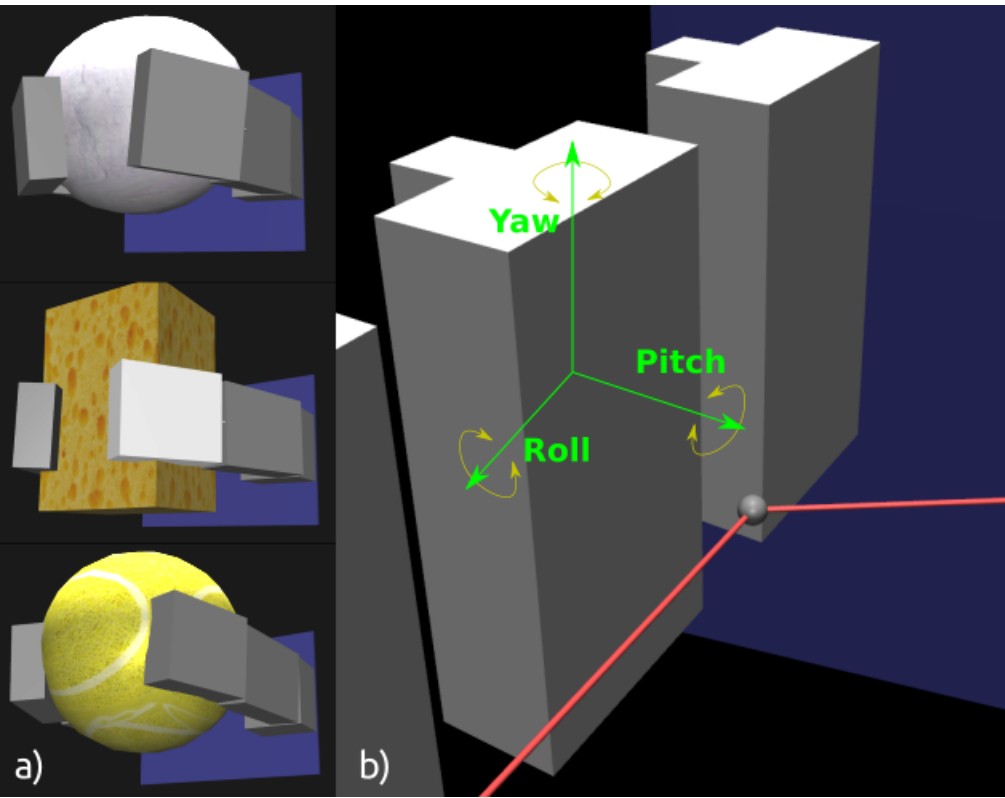

**Figure 4.** Soft-robotic gripper in the MuJoCo environment: (**a**) the gripper squeezes and releases objects in three shapes—a ball, a box and a cylinder, all with a variable stiffness parameter; (**b**) each geometrical block of each finger is connected to others by three hinges. In this setup, we can easily adjust the ranges of each joint in a roll, pitch and yaw axes.

A stiffness coefficient in all our experiments (including the real-world scenario) is defined in the same way as in the MuJoCo simulator—the *softness* of an object is characterised

as the stiffness of springs attached from one side to the geometrical blocks on a surface and from the other side to the centre of it. We always assume that the object is homogeneous.

The data collection was performed using the following steps. An object is located between fingers and the actuator starts to close the gripper to squeeze the object. After a half of an episode, the gripper opens. During the process, an object is embraced by fingers that adapt themselves to its shape. A stiffness coefficient is expressed in N/m and varies among episodes to equally cover the range (from 300 to $1400\frac{N}{m}$), which fits the real-world data range. A mass of all parts was adapted to the real values, as well as the mechanical impedance of objects, damping, and stiffness of all joints and springs in the system. Two IMUs are mounted on a MuJoCo's element called *site* and located in the 3/4 of the length of each finger in the outside part of it. For experiments, we prepared two simulation datasets. The first one resembles the real-world data and consists of 5000 training-validation samples gathered from squeezing the box object only. We use it for an enrichment of real-world data. The second one was composed of objects in three different shapes—boxes, cylinders and spheres. It counts 3999 training-validation samples—1333 samples per each object. In our research, it was used to verify whether the NN can avoid overfitting to any particular shape. Additionally, to verify the NN performance among different shapes of objects we prepared three test datasets—133 samples for each object.

### 3.3. Experimental Design

The performance of Neural Networks (NN) was verified using a k-fold cross-validation technique in each experiment. That method assesses the error rate and the generalisation ability of predictive models. In our research, data is processed as follows: we shuffle the dataset, then split the dataset into k subsets (folds), proceed with training using the $k-1$ folds of data and validate the performance at the end of an epoch using the k-th fold. Additionally, unless otherwise stated, after each epoch, we test the current NN model using separate test data. After that, the procedure is repeated by starting the training of a neural network from scratch on other folds of data. In our research, to ensure a fair comparison of trained NNs, we did the 5-fold cross-validation for all experiments. As we perform the regression task, we chose a Mean Absolute Error (MAE) and a Mean Absolute Percentage Error (MAPE) as the performance metrics, to verify both absolute and relative errors. Considering the usage of the cross-validation technique, in the following description of datasets, we provided the number of samples in the training-validation sets together and separately for test sets if needed. The summary of all datasets used in our experiments was presented in Table 2.

**Table 2.** The number of samples in datasets used in our experiments based on the cross-validation.

| Name | Train/Validation | Test |
|---|---|---|
| Simulation (box only) | 5000 | - |
| Simulation (all shapes) | 3999 | 399 |
| Real-world | 200 | 300 |

### 3.4. Network Architecture

Our neural networks were predicting the stiffness parameter from fixed length sequences of accelerations and angular velocities measured by IMUs. In our research we proposed to test three types of neural networks—the ConvNet based entirely on 1D convolutional blocks, the ConvLSTMNet with forward LSTM units and the ConvBiLSTMNet with bidirectional LSTM units. In both cases of LSTM-based NNs models, the recurrent part is placed after the convolutional block. At the end of each architecture, we placed a fully-connected layer named the Regression Block. The scheme of proposed neural network architectures are depicted in Figure 5.

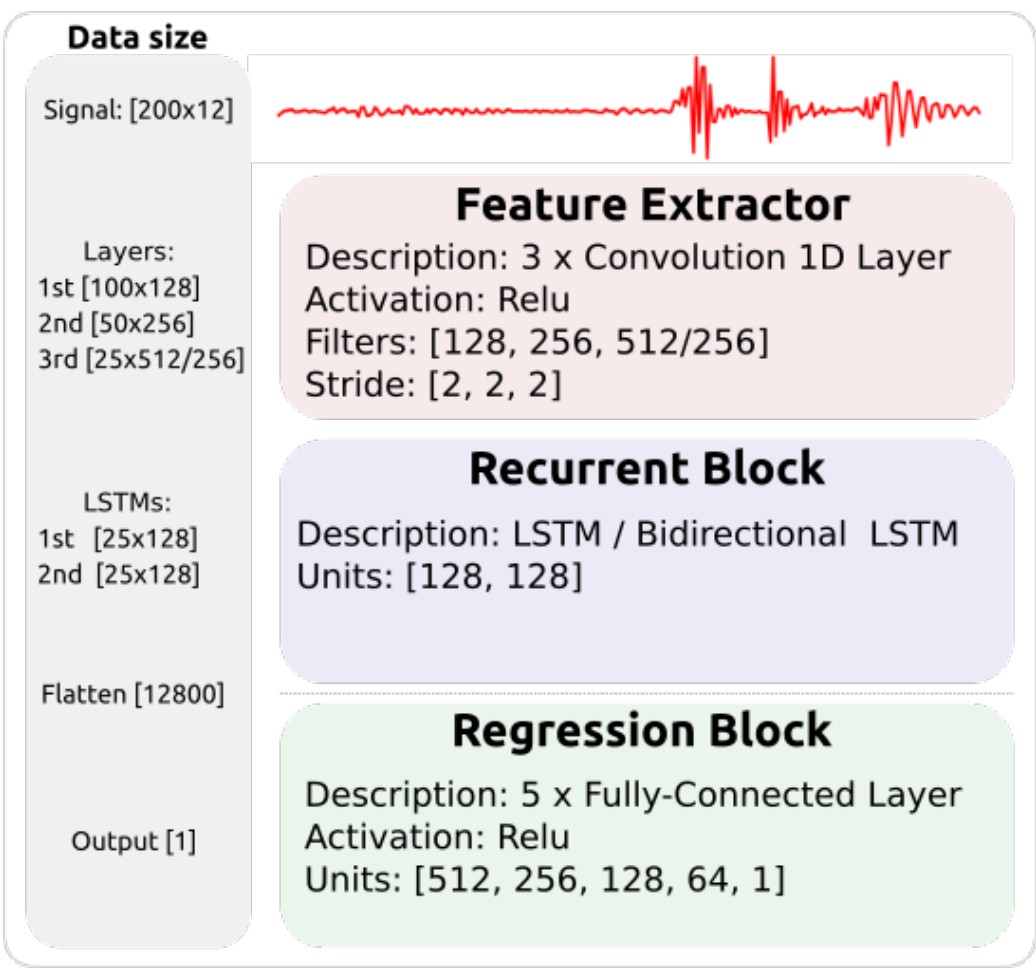

**Figure 5.** In our networks, the Feature Extractor produced high-level features from an input using 1D convolutions. In the ConvBiLSTMNet and the ConvLSTMNet, the Recurrent Block processed these features to find relevant connections for the stiffness estimation. However, in the former architecture it was done in the forward and backward manner (from the beginning of the signal and back). Finally, the Regression Block transformed high-level features into one scalar value. In our experiments, we exploited three architectures of neural networks. The difference is in the Recurrent Block—both recurrent NNs have the reduced number of filters in the last convolutional layer and added LSTM cells with 256 units ($2 \times 128$), while in the ConvNet the output from the Feature Extractor is passed directly to the Regression Block.

Feature Extractor—The neural network input was a standardised sensor reading in the form of the two-dimensional tensor. Each sample consisted of 12 time series with a length of 200. The main task of that block is to extract features while remaining in the time domain. Hence, data could be further processed recurrently or passed to the Regression Block directly. The Feature Extractor consisted of three consecutive 1D convolution layers with strides equal 2. In the ConvNet the number of filters was set to 128, 256, 512, while in the ConvLSTMNet/ConvBiLSTMNet, the last convolution block was reduced to 256 filters and replaced by the recurrent block with the same size.

Recurrent Block—It processes high dimensional time series from the Feature Extractor in a recurrent manner using LSTM [46] or bidirectional LSTM cells [47]. The input is mapped to a fixed-length vector that represents the entire sensor reading in itself. In that way, we obtained a global, reduced description of the signal. Each recurrent cell consists of 128 units, as depicted in Figure 6. In the the ConvLSTMNet, both LSTM cells are organised in two sequential layers processing the input in the forward direction only. Outputs of that block are finally forwarded to the Regression Block.

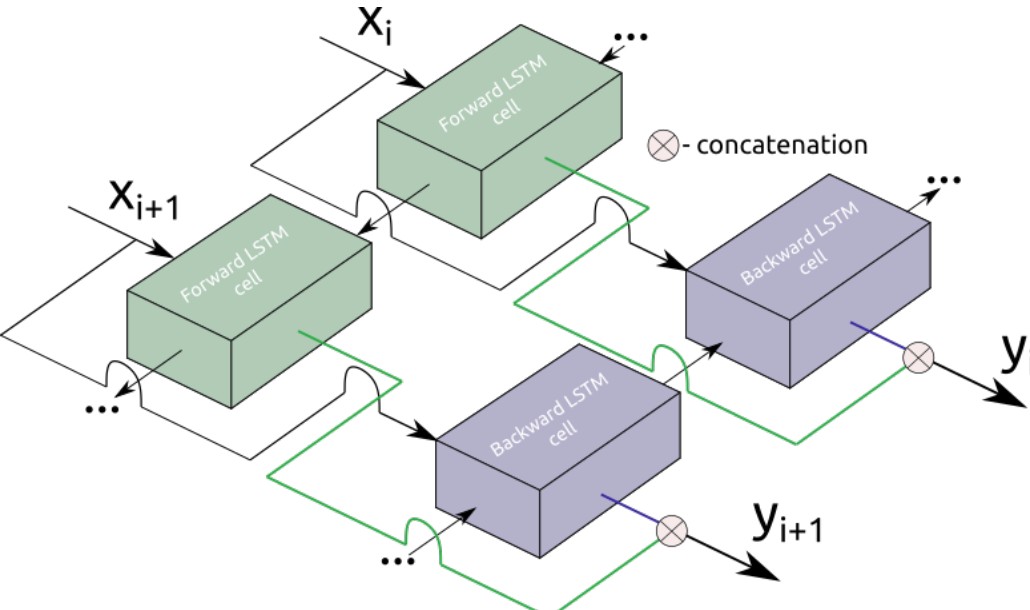

**Figure 6.** The core idea standing behind the bidirectional LSTM used in the ConvBiLSTMNet is as follows–to prevent losing a context by the cell, process a sequence from the beginning to the end, do the same in the reversed direction and concatenate both *passes*. Input $x_i$ refers to the *i*-th feature vector returned by the convolutional block.

Regression Block—The last block was used to do a regression and output an estimated stiffness coefficient. The necessity of using a fully-connected block stems from the fact that extracted features and time dependencies between them are critical ingredients in the regression process, but they are not the answer itself. At the very end of the processing, it is necessary to transform the obtained features into stiffness coefficient estimate, which can be easily performed using the stack of fully-connected layers. The number of units in each layer remains unchanged for all tested architectures and is 512, 256, 128, 64, 1.

## 4. Results

The section of the results was divided as follows. Firstly, the simulation results were presented. We verified which NN yielded the best performance on the simulation datasets and how well it was able to generalise among different shapes of squeezed objects. After that, the real-world experiments were conducted using the NN architecture chosen during the simulation test stage. Finally, we focused on the closing of the reality gap between simulation and real-world data. In all experiments we validated or models using the k-fold cross-validation technique and provided results obtained for the best epoch per each fold according to the MAPE. To ensure a fair comparison, in all our experiments and tests we used Adam optimiser with a learning rate set to 0.001. Each model was trained with the batch size 100 and all our solutions were trained for 100 epochs per each fold of the cross-validation.

### 4.1. Neural Network Architecture Comparison

First of all, in our experiments, we compared three types of neural networks using data from simulation and chose the best one for further experiments. The values of MAE/MAPE metrics from the cross-validation procedure were presented in Table 3. The best performing network—the ConvBiLSTMNet, was chosen for further experiments.

**Table 3.** The comparison of three NN architectures according to MAE/MAPE metrics. The usage of bidirectional LSTM units gave an improved performance comparing to the ConvNet and the ConvLSTMNet .

|        | ConvNet | | ConvLSTMNet | | ConvBiLSTMNet | |
|--------|------|------|------|------|------|------|
| **k-Fold** | **MAE** | **MAPE** | **MAE** | **MAPE** | **MAE** | **MAPE** |
| I    | 19.1 | 2.4 | 6.2 | 0.8 | 6.2 | 0.8 |
| II   | 11.8 | 1.6 | 5.4 | 0.7 | 5.4 | 0.7 |
| III  | 15.1 | 2.2 | 7.8 | 1.1 | 7.8 | 1.1 |
| IV   | 14.6 | 1.9 | 6.7 | 0.9 | 6.7 | 0.9 |
| V    | 18.1 | 2.1 | 6.2 | 1.0 | 6.2 | 1.0 |
| **MEAN** | 15.7 | 2.0 | 6.8 | 0.9 | 6.5 | 0.9 |
| **SD**   | 2.9  | 0.3 | 0.9 | 0.2 | 0.7 | 0.1 |

*4.2. Shape Generalisation*

To verify the capability of the ConvBiLSTMNet to successfully estimate the stiffness parameter we conducted more experiments using the simulation-only datasets. We started the cross-validation procedure from scratch for chosen model and reported the MAE/MAPE for three different datasets in Table 4. Each test dataset was composed of sensor readings from squeezing only one type of object so that the findings of the shape-dependent stiffness parameter regression could be provided.

**Table 4.** The results from experiments on shape-invariant estimation of the stiffness parameter using ConvBiLSTMNet.

|        | Dataset | | | | | |
|--------|------|------|------|------|------|------|
| **k-Fold** | **Ball** | | **Box** | | **Cylinder** | |
|        | **MAE** | **MAPE** | **MAE** | **MAPE** | **MAE** | **MAPE** |
| I    | 20.3 | 2.0 | 24.1 | 1.8  | 15.6 | 1.8 |
| II   | 29.6 | 2.6 | 12.9 | 1.6  | 15.8 | 1.9 |
| III  | 27.1 | 2.0 | 22.8 | 1.8  | 16.0 | 1.9 |
| IV   | 21.8 | 2.1 | 17.7 | 16.6 | 18.4 | 1.9 |
| V    | 19.3 | 2.0 | 24.4 | 1.5  | 20.8 | 1.9 |
| **MEAN** | 23.6 | 2.1 | 20.4 | 4.7  | 17.3 | 1.9 |
| **SD**   | 4.5  | 0.3 | 5.0  | 6.7  | 2.2  | 0.0 |

*4.3. Sim-To-Real Gap*

The central part of our research was about assessing the reality gap in the task of the stiffness parameter estimation. In that part of the experiments, we performed 5 training procedures of the ConvBiLSTMNet on data with different number of real-life examples or noise added to simulation data, each composed of 5-fold cross-validation. In Table 5 we reported MAE/MAPE metrics gathered while *testing* each model on the separate dataset, not involved in the training/validation procedure. In the *sim + noise* experiment we tried to close the reality gap, by adding a zero-mean Gaussian noise with standard deviation set to $0.7 \frac{m}{s^2}$ for accelerations and $0.06 \frac{rad}{s}$ for the gyroscope readings. The parameters of the noise were adjusted by trials and errors, thus too large standard deviation resulted in the lack of the convergence ability of the NN, while too small caused model to overfit to the simulation data and no clear rule for that phenomena is known. Each next cross-validation turn was performed on simulation datasets without noise and with a small number N of real-world data samples included in the training part. In Table 5 we refer to them as *sim + N real*.

**Table 5.** MAE/MAPE results reported for best epochs from each of the cross-validation turns. Introducing to the network even a small number of real-world sensor readings resulted in a significant improvement in the performance.

| Experiment Name | k-Fold | | | | | | | | | | MEAN | |
| | I | | II | | III | | IV | | V | | | |
| | MAE | MAPE | MAE | MAPE | MAE | MAPE | MAE | MAPE | MAE | MAPE | MAE | MAPE |
|---|---|---|---|---|---|---|---|---|---|---|---|---|
| sim + noise | 281.3 | 37.7 | 275.0 | 38.5 | 275.6 | 38.4 | 282.7 | 37.6 | 256.6 | 37.9 | $274.2 \pm 10.4$ | $38.0 \pm 0.4$ |
| sim + 50 real | 190.6 | 23.1 | 216.1 | 27.1 | 187.8 | 26.4 | 151.8 | 21.6 | 200.7 | 27.7 | $189.4 \pm 23.8$ | $25.2 \pm 2.7$ |
| sim + 100 real | 134.6 | 20.6 | 108.3 | 17.6 | 134.9 | 19.6 | 126.8 | 18.6 | 126.6 | 18.3 | $126.2 \pm 10.8$ | $18.9 \pm 1.2$ |
| sim + 150 real | 89.3 | 12.9 | 85.9 | 13.7 | 92.7 | 13.2 | 73.9 | 11.0 | 79.9 | 10.2 | $84.3 \pm 7.5$ | $12.2 \pm 1.5$ |
| sim + 200 real | 66.9 | 9.1 | 49.3 | 7.0 | 82.6 | 10.9 | 67.4 | 8.4 | 56.6 | 8.0 | $64.6 \pm 12.6$ | $8.7 \pm 1.5$ |

## 5. Discussion

In the following section, we summarised obtained results and our observations for three types of experiments carried out in the course of our research.

Architecture Choice—We compared the performance of three types of neural networks in the task of a stiffness parameter estimation from IMUs readings, to choose the best one for the further analysis. All models were examined on the simulation dataset without real-world data samples. In Table 3 one can observe the results from cross-validation on the simulation dataset. The mean results of the MAE/MAPE show the advantage of the LSTM-based models in the performed task. The conclusions are twofold. Firstly, the ConvBiLSTMNet is more accurate in its predictions than ConvNet, resulting in MAE of $6.5\frac{N}{m}$ and MAPE of 0.9%,which means the improvement over $9.5\frac{N}{m}$ and 1.1% achieved by the ConvNet. Secondly, the stability of the learning process also improved and it can be observed in deviations of errors obtained between cross-validation folds. For ConvNet the standard deviation of results is $2.9\frac{N}{m}$ MAE and 0.3% MAPE, while the ConvBiLSTMNet decreased these values to $0.9\frac{N}{m}$ and 0.2% respectively. Comparing two recurrent NNs, one can observe that the results are similar. However, the ConvBiLSTMNet exhibits better performance in the MAE, what means than on average it made a lesser absolute error, hence that architecture was chosen for further experiments.

Shape-Invariant Predictions—To verify the generalisation capability of the ConvBiL-STMNet and verify its performance on different types of objects, we performed additional experiments. In Table 4 we gathered the MAE/MAPE from testing the network on three separate datasets, each of which included only one shape of object, while training on all shapes at once. All the results suggest that the proposed NN was able to generalise among different types of shapes and perform the shape-invariant stiffness parameter prediction. It appears that the cylinder-shaped objects are the easiest in the performed task, which is reflected in the lowest errors $17.3\frac{N}{m}$ MAE and 1.9% MAPE. However, box objects gave the smaller values of MAE ($20.4\frac{N}{m}$) than ball-shaped objects ($23.6\frac{N}{m}$), while looking at the MAPE the situation was the opposite—larger error was observed for boxes (4.7%/2.1%). This means that the NN was inaccurate more often while estimating large stiffness values for boxes that resulted in the increased relative metric (MAPE), while for ball-shaped objects the quality of the estimation was decreased for small values that gave increased absolute measure (MAE).

Closing The Reality Gap—In the task of haptic recognition of physical parameters, data from the physics simulator appeared to resemble the real-world IMU readings only to some restricted extent. Although the results from *sim + noise* tests were significantly worse than any of the *sim + real* trail, the mean MAPE 38% suggests that the correspondence between the simulation-only and real-world signals exists. Additionally, it is important to note that MAE/MAPE values from each fold in the *sim + noise* experiment remained relatively close to each other, which means that the model prediction performance was similar for the entire dataset, as it was equally balanced in the stiffness parameters range. However, the reality gap cannot be considered as a solved problem, because the greatest

improvement was observed for experiments with the real-world sensor readings included in the training dataset. In Figure 7, one can observe the decreasing value of MAE/MAPE metrics as the number of real data samples are added to the training dataset. In our experiments we do not include the results from the training on the real-world data only, as they would be incomparable with other experiments, due to the low variability of the stiffness coefficient. Additionally, the number of data samples would be too low to assess the fair comparison in the real-world scenario. The lowest MAE/MAPE obtained in experiments on closing the reality gap were achieved for *sim + 200 real* trial and were equal to $64.6\frac{N}{m}$ and 8.7%. However, in the *sim + 50 real* experiment, the added number of real samples constituted only 1.2% of the entire training dataset, but the largest performance improvement among all experiments was observed. The improvement was $84.8\frac{N}{m}$ and 12.8% of the MAE/MAPE.

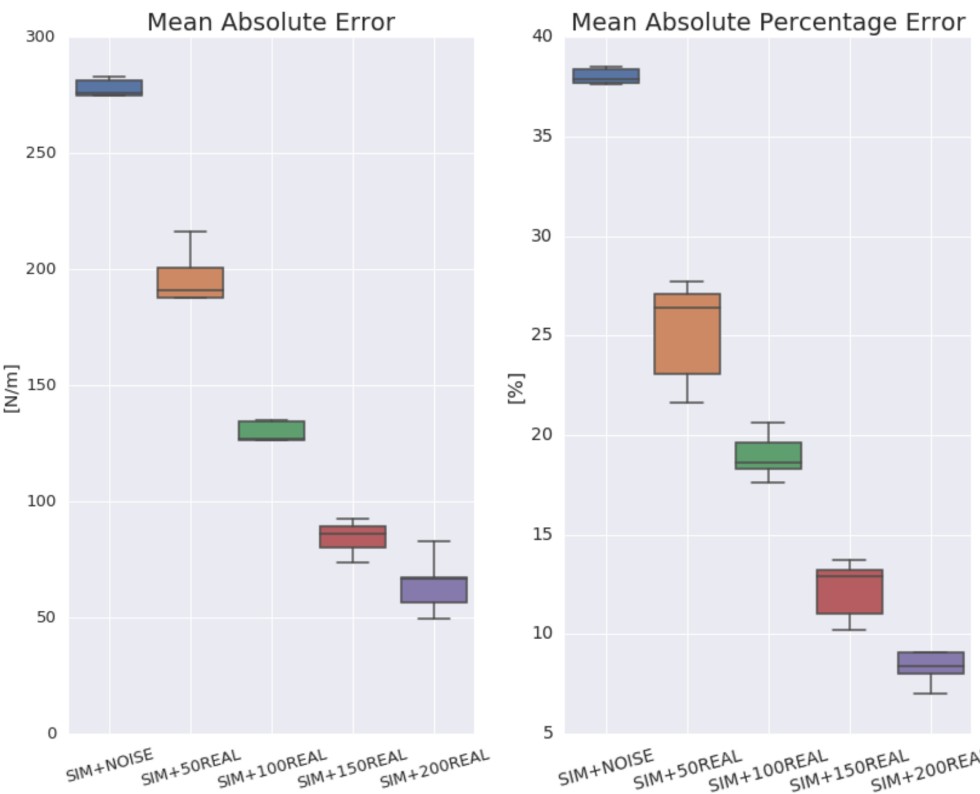

**Figure 7.** Results of MAE/MAPE from the testing on real-world data presented in the box plot. As the number of real data samples included in the training dataset increases, the test error decreases. Boxes represent consecutive experiments and consist of the five-number summary of the result (from the bottom of each box): minimum, first quartile, median, third quartile and maximum value.

## 6. Conclusions

We have shown that estimation of the object's physical parameters using data from IMU sensors is possible and beneficial due to the low cost of setup and no further need for sophisticated equipment. Our deep learning solution solves a problem of the stiffness estimation in the soft robotics area, introducing a novel approach, which associates an embodied and artificial intelligence. Their combination may lead to a system robust to unforeseen and changing external conditions. While currently used methods of stiffness search exploit techniques of measurement or direct estimation, the method proposed by us is characterised by the discovery of knowledge and causal relationships related to the characteristics of a given object and its physical features. Research on the discovery of knowledge acquired by neural networks may result in the diagnosis of the intuition behind the natural behaviour of humans in the tasks of manipulating objects. We find it likely that similar solutions, based on low-cost sensors and deep learning, may be successfully

applied for robotic manipulation in everyday scenarios. We hope that the published data and the implementation of neural networks used in our experiments will inspire other researchers to delve into the research area of soft grippers and perception of the physical world based on tactile data in robotics.

**Author Contributions:** Conceptualization, M.B. and J.B.; Formal analysis, P.K.; Funding acquisition, K.W.; Methodology, M.B.; Software, M.B.; Supervision, P.K. and K.W.; Writing—original draft, M.B., P.K., J.B. and K.W.; Writing—review and editing, M.B. and K.W. All authors have read and agreed to the published version of the manuscript.

**Funding:** This work is supported by grant No. LIDER/3/0183/L-7/15/NCBR/2016 funded by The National Centre for Research and Development (Poland).

**Conflicts of Interest:** The authors declare no conflict of interest.

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
