# Peer review of "Gaining a Sense of Touch Object Stiffness Estimation Using a Soft Gripper and Neural Networks"

_electronics, doi:10.3390/electronics10010096_

Round 1
Reviewer 1 Report
Manuscript Number: electronics-1041600
Title: GAINING A SENSE OF TOUCH. OBJECT STIFFNESS ESTIMATION USING A SOFT GRIPPER AND NEURAL NETWORKS
Generally, this multidisciplinary study presents an interesting application of soft robotics. However, the manuscript consists of weaknesses which must be corrected.
*My main question is about the applicability of the soft gripper studied in this paper. More precisely, we know that a soft robotic gripping system enables industrial and collaborative robots to adaptively handle unstructured tasks and objects of varying size, shape and weight. I can see some notes about Universal Robot UR3 collaborative manipulator in Page 5. However, I would like to know how easy is to integrate this soft gripper to robots which are already used in automating applications like:
- Primary and secondary food handling and packaging
- High-speed pick and place lines
- Consumer packaged goods
- Assembly
- Machine tending
- CNC machine automation
Does the soft gripper enable automation in new markets and make it easy for all users to validate solutions quickly? The answer can be related to both cost and technologies advantages. The answer can be also related to the flexibility of grippers (e.g., more flexible connection mechanisms in each robotic module such that each cell could connect to many other modules.)
*Sections should be numbered as they start from 1. Accordingly, instead of “0. Introduction, 1. Related work, 2. Method, etc.”, the heading of sections should be “1. Introduction, 2. Related work, 3. Method, etc.”.
*The number of typo errors in this paper is quite high, and consequently it benefits from a comprehensive check. I just give some examples:
-Caption of Figure 1: set of defromable --> set of deformable
-Caption of Figure 1: object stifness cooficent regression --> object stiffness coefficient regression
-Page 2, Line 40: Our contribution are --> Our contributions are
-Page 2, Line 57: set of deformable object --> set of deformable objects
-Page 3, Line 66: In Figure 2 there was --> In Figure 2, there is
-Page 5, Line 147: e.g. the greater --> e.g., the greater
-Page 6, Line 182: As it was depicted --> As it is depicted
-Page 6, Line 186: as was depicted in --> as is depicted in
-Page 8, Line 228: architectures was depicted --> architectures is depicted
-Page 12, Line 320: In Figure 7 one can --> In Figure 7, one can
-Page 13, Line 350: Grasping With Underactuated --> Grasping with Underactuated
-Page 15, Line 408: Trans. Sig. Proc. --> IEEE Transactions on Signal Processing
*Although the main focus is on robotic grippers, I believe this paper does not have a general literature review in the sense of grippers. There are many recent innovative gripper designs for industries. Specifically, hybridization of grippers functions is well studied and accordingly the following papers should be cited in the first section to show the generality of the study of robotic grippers to interested readers.
[a] Design and implementation of a multi-function gripper for grasping general objects, Applied Sciences MDPI, 2019, vol. 9, no. 24, 5266
[b] Notes on feasibility and optimality conditions of small-scale multifunction robotic cell scheduling problems with pickup restrictions, IEEE Transactions on Industrial Informatics, 2015, vol.11, no.3, pp. 821 - 829
*Figure 7 is in the conclusion section, but I believe that it should be moved on top of previous page to be inside Section 4.
*There is an inconsistency in the reference list in Pages 13 and 14. The titles of some papers are “capitalized each word”, but not all. Please make all of them same.
*The future research directions are not directly mentioned in the conclusion section. Please do not write about hopes (We hope that the published data … on tactile data in robotics.). Instead, imply one or two attractive future research directions.
Author Response
Dear Reviewers,
thank you for your time spent reading our paper and providing us with comments and suggestions on improving our article. We carefully read the reviews and prepared the answers in this document. Additionally, the changes done in the manuscript are marked with the color.
Generally, this multidisciplinary study presents an interesting
application of soft robotics. However, the manuscript consists of
weaknesses which must be corrected.
*My main question is about the applicability of the soft gripper
studied in this paper. More precisely, we know that a soft robotic
gripping system enables industrial and collaborative robots to
adaptively handle unstructured tasks and objects of varying size,
shape and weight. I can see some notes about Universal Robot
UR3 collaborative manipulator in Page 5. However, I would like
to know how easy is to integrate this soft gripper to robots which
are already used in automating applications like:
Primary and secondary food handling and packaging
High-speed pick and place lines
Consumer packaged goods
Assembly
Machine tending
CNC machine automation
Response:
The application of soft grippers is easy to integrate for all the applications where precision is not of great concern. Therefore I could easily imagine the use of Primary and secondary food handling and packaging as it is the case for the projects done for Ocado in the UK within the framework of the project SoMa: http://soma-project.eu/.
However, in cases where high precision is required, the soft grippers are not ready to be integrated into the automation process.
The soft gripper integration is still not easy as they have poor position control of the fingers as the classical approach with the joint encoders can not be used due to the lack of classical joints.
*Does the soft gripper enable automation in new markets and
make it easy for all users to validate solutions quickly? The
answer can be related to both cost and technologies
advantages. The answer can be also related to the flexibility of
grippers (e.g., more flexible connection mechanisms in each
robotic module such that each cell could connect to many other
modules.)
Response:
Indeed, the soft grippers are opening new markets as they allow to change how the objects are grasped. In classical robotics manipulation, the grippers are rigid; hence, the fingers' contact is avoided not to damage the gripper. With the soft grippers, we can touch the environment and exploit additional contacts. The new markets are envisioned to be the one which requires direct and safe collaboration with the human. There are currently many options for buying a cobot on the market, but there is a low number of grippers that secure safe collaboration with humans.
*Sections should be numbered as they start from 1. Accordingly,
instead of “0. Introduction, 1. Related work, 2. Method, etc.”, the
heading of sections should be “1. Introduction, 2. Related work,
- Method, etc.”.
Response: Corrected.
*The number of typo errors in this paper is quite high, and
consequently it benefits from a comprehensive check. I just give
some examples:
-Caption of Figure 1: set of defromable --> set of deformable
Response: Corrected.
-Caption of Figure 1: object stifness cooficent regression -->
object stiffness coefficient regression
Response: Corrected.
-Page 2, Line 40: Our contribution are --> Our contributions are
Response: Corrected.
-Page 2, Line 57: set of deformable object --> set of deformable
objects
Response: Corrected.
-Page 3, Line 66: In Figure 2 there was --> In Figure 2, there is
Response: Corrected.
-Page 5, Line 147: e.g. the greater --> e.g., the greater
Response: Corrected.
-Page 6, Line 182: As it was depicted --> As it is depicted
Response: Corrected.
-Page 6, Line 186: as was depicted in --> as is depicted in
Response: Corrected.
-Page 8, Line 228: architectures was depicted --> architectures
is depicted
Response: Corrected.
-Page 12, Line 320: In Figure 7 one can --> In Figure 7, one can
Response: Corrected.
-Page 13, Line 350: Grasping With Underactuated --> Grasping
with Underactuated
Response: Corrected.
-Page 15, Line 408: Trans. Sig. Proc. --> IEEE Transactions on
Signal Processing
Response: Corrected.
*Although the main focus is on robotic grippers, I believe this
paper does not have a general literature review in the sense of
grippers. There are many recent innovative gripper designs for
industries. Specifically, hybridization of grippers functions is well
studied and accordingly the following papers should be cited in
the first section to show the generality of the study of robotic
grippers to interested readers.
[a] Design and implementation of a multi-function gripper for
grasping general objects, Applied Sciences MDPI, 2019, vol. 9,
- 24, 5266
[b] Notes on feasibility and optimality conditions of small-scale
multifunction robotic cell scheduling problems with pickup
restrictions, IEEE Transactions on Industrial Informatics, 2015,
vol.11, no.3, pp. 821 - 829
Response:
The whole section of the related work was added in the text. The papers [a] and [b] are also included in the text.
*Figure 7 is in the conclusion section, but I believe that it should
be moved on top of previous page to be inside Section 4.
Response: Corrected.
*There is an inconsistency in the reference list in Pages 13 and
- The titles of some papers are “capitalized each word”, but
not all. Please make all of them same.
Response: Corrected.

Reviewer 2 Report
Estimating and predicting stiffness in real-time gripping are arousing wide interest in the design and control of soft actuators. This works demonstrate three different neural network architectures comparatively and the discoveries are of value to following attempts in this methodology. On a higher level of overview, however, more readings should be added into the Introduction part to provide a wider view of the emerging AI-Actuators fusion.
Some examples are (1)"Chin et al, Machine Learning for Soft Robotic Sensing and Control, 2020", (2)"Al-lbadi et al, Front. Robot. AI, 05 October 2020 | https://doi.org/10.3389/frobt.2020.00115". A brief review on similar works are encouraged to indicate how various structures of NNs affect simulation/estimation accuracy and efficiency.
The current work builds a solid database on "estimated stiffness vs measure stiffness" for each NN, but don't tell how "accurate prediction of stiffness" achieved here improves practical use of a soft gripper. Such works should be considered to complete the whole picture.
Author Response
Dear Reviewers,
thank you for your time spent reading our paper and providing us with comments and suggestions on improving our article. We carefully read the reviews and prepared the answers in this document. Additionally, the changes done in the manuscript are marked with the color.
Estimating and predicting stiffness in real-time gripping are
arousing wide interest in the design and control of soft actuators.
This works demonstrate three different neural network
architectures comparatively and the discoveries are of value to
following attempts in this methodology. On a higher level of
overview, however, more readings should be added into the
Some examples are (1)"Chin et al, Machine Learning for Soft
Robotic Sensing and Control, 2020", (2)"Al-lbadi et al, Front.
Robot. AI, 05 October 2020
| https://doi.org/10.3389/frobt.2020.00115". A brief review on
similar works are encouraged to indicate how various structures
of NNs affect simulation/estimation accuracy and efficiency.
Response:
The paper was added to the state of the art section. Together with a brief review of similar works. Marked with green in the text.
The current work builds a solid database on "estimated stiffness
vs measure stiffness" for each NN, but don't tell how "accurate
prediction of stiffness" achieved here improves practical use of a
soft gripper. Such works should be considered to complete the
whole picture.
Response:
We agree that the work on the practical use of obtained information about the object stiffness in soft gripper use is a valuable contribution. However, our paper focuses on perception, not on finger motion control and path planning. Hence, the work on the practical use of stiffness parameters might be a good follow up paper, but it is out of the current article's scope.

Round 2
Reviewer 1 Report
The paper is worth publishing in the current format. My comments are mostly applied in the current revision. So, it can be accepted as it is.